# Diabetes Mellitus and Other Predictors for the Successful Treatment of Metastatic Colorectal Cancer: A Retrospective Study

**DOI:** 10.3390/medicina58070872

**Published:** 2022-06-29

**Authors:** Deiana Roman, Sorin Saftescu, Bogdan Timar, Vlad Avram, Adina Braha, Șerban Negru, Andrei Bercea, Monica Serbulescu, Dorel Popovici, Romulus Timar

**Affiliations:** 1Second Department of Internal Medicine, “Victor Babeș” University of Medicine and Pharmacy, 300041 Timisoara, Romania; roman.deiana@umft.ro (D.R.); timar.bogdan@umft.ro (B.T.); avram.vlad@umft.ro (V.A.); braha.adina@umft.ro (A.B.); timar.romulus@umft.ro (R.T.); 2Department of Oncology, “Victor Babeș” University of Medicine and Pharmacy, 300041 Timisoara, Romania; snegru@yahoo.com (Ș.N.); dorel.popovici@umft.ro (D.P.); 3OncoHelp Hospital, 300239 Timisoara, Romania; bercea.andrei@gmail.com (A.B.); serbulescu.monica@yahoo.com (M.S.)

**Keywords:** diabetes mellitus, colorectal cancer, metastatic cancer, chemotherapy

## Abstract

*Background and Objectives*: In the last decades there has been an increasing body of research identifying the positive correlation between diabetes mellitus (DM) and solid malignancies, moreover, having shown DM as an independent risk factor for colorectal cancer (CRC). The aim of the present study was to assess the impact of DM on metastatic CRC (mCRC), and to identify possible predictive factors in the successful treatment of mCRC. *Materials and Methods*: 468 patients with mCRC were included in this retrospective, observational study. A total of 8669 oncological treatment cycles related to 988 distinct chemotherapy lines were analyzed. Data regarding lines of treatment and blood panel values were obtained from the Oncohelp Hospital database. *Results*: The presence of DM in male patients >70 years was a negative predictor (RR = 1.66 and a *p* = 0.05). DM seemed to have a detrimental effect in patients whose treatment included bevacizumab (median time to treatment failure -TTF- 94 days for DM+ cases compared to 114 days for DM-patients, *p* = 0.07). Analysis of treatments including bevacizumab based on DM status revealed lower values of mean TTF in DM+ female patients versus DM-(81.08 days versus 193.09 days, *p* < 0.001). It was also observed that DM+ patients had a higher mean TTF when undergoing anti-EGFR (epidermal growth factor) therapy (median TTF 143 days for DM+ patients versus 97.5 days for those without DM, *p* = 0.06). *Conclusions*: The favorable predictive factors identified were the inclusion of antiangiogenic agents, a higher hemoglobin value, a higher lymphocyte count, the inclusion of anti-EGFR treatment for DM+ patients, a higher creatinine, and a higher lymphocyte count in treatment lines that included anti-EGFR treatment. Unfavorable predictive factors were represented by the presence of DM in female patients undergoing antiangiogenic treatment, neutropenia in male patients, the association of oxaliplatin and antiangiogenic agents, and a higher monocyte count in the aforementioned treatment lines.

## 1. Introduction

Colorectal cancer (CRC) remains among the most frequent malignancies throughout the world. In terms of mortality, it stands in the second position, while in terms of incidence it occupies the third [1]. In 2020, as reported by Globocan, 1,931,590 estimated new cases of colorectal cancer were recorded globally, being the second most common malignancy in women and the third most diagnosed cancer in men. The lifetime risk of developing colorectal cancer has been approximated to be 1 in every 25 women (4.0%), slightly lower than that of the male gender (4.3%), equivalent to 1 in every 23 [2].

CRC has been recognized as a marker of the level of social and economic development of a country, with research suggesting a substantial increase in CRC incidence and mortality is to be observed in nations with a medium or high human development index (HDI), due to the adherence to a western dietary pattern and lifestyle [3]. It has been observed that the incidence and mortality of CRC also vary between both race and ethnicity, with non-Hispanic black people of color having the highest rates, while Asian Americans/Pacific Islanders have the lowest. Through early detection, as well as early removal of precancerous adenomas, extensive CRC screening has significantly decreased incidence and mortality among all risk groups [4].

There is a growing body of research underlining the importance of dietary and lifestyle variables influencing the risk of colorectal cancer. Physical activity appears to have a protective effect, according to multiple studies. A higher frequency of red and processed meat consumption, as well as alcohol intake, increase the risk of developing CRC [5]. The effects of different compounds found in meat and meat products, such as animal protein, heme, N-nitroso compounds, and heterocyclic amines on the mucosa of the gastrointestinal tract, as well as on the gut microbiome can cause genotoxicity and metabolic abnormalities [6]. Moreover, the same microbiome is modified by frequent antibiotic use [4]. A better understanding of these elements could have significant public health implications, particularly given the alarming velocity at which new cases are rising [7].

A steady increase in new cases globally is also seen in Diabetes Mellitus (DM), according to the 2021 Diabetes Atlas elaborated by the International Diabetes Federation (IDF), with roughly 537 million adults aged between 20 and 79 years old diagnosed worldwide, this number being expected to increase to more than 630 million individuals by the year 2030 [8].

As early as 1910, studies have been conducted in an attempt to find a positive correlation between cancer in all its forms and DM. However, while some studies such as the one led by Doering et al. reported a positive correlation, others like the one led by Stevenson et al. in 1921 showed that no correlation was to be found. In conclusion, studies have failed to unequivocally prove that diabetes is a predisposing factor for cancer, as stated in “Diabetes and Cancer”, written by Elliott P.

During the last decades, research has found, through numerous published studies and meta-analyses, that there is a positive correlation between diabetes and some solid malignancies such as hepatocarcinoma and renal cell cancer [9,10]. Worldwide, studies have identified DM as a significant risk factor for CRC development [11]. For example, a meta-analysis involving six case-control and nine cohort studies conducted between 1966 and 2005, including approximately 2.6 million participants, arrived at the conclusion that diabetes was associated with an increased risk of colorectal cancer. The meta-analysis result concluded that diabetes increases the relative risk of developing a colorectal malignancy by approximately 30% [12]. One of the hypotheses for arriving at such a conclusion could be that there is a commonality in risk factors between the two pathologies, such as lack of physical activity, obesity, and advancement in age. In addition, hyperinsulinemia, or factors that promote insulin resistance, hyperglycemia, and hypertriglyceridemia, are also present in carcinogenesis. Diabetes mellitus is one of the leading causes of mortality worldwide. Together with cardiovascular disease, cancer, and respiratory disease, these pathologies account for more than 80% of all premature non-communicable disease (NCD) fatalities [3]. Patients with DM present a two- to three-fold increase in all-cause mortality risk [4]. DM increases the risk of acute complications, such as infections, as well as that of chronic complications, such as cardiovascular disease, stroke, chronic kidney disease, chronic liver disease, and cancer mortality [6,13]. Furthermore, despite strenuous efforts being made for improving population health and life expectancy, diabetes has the second-largest negative total effect on health-adjusted life expectancy globally [14].

According to Ana Silva et al., obesity is likely to have a negative impact on CRC outcomes, particularly in male patients [13]. The tumor microenvironment includes adipocytes, which have in turn been shown to be biologically programmed to become highly catabolic, thus resulting in the production of free fatty acids, leading to an accelerated tumor growth and migration [15].

A key determinant involved in both the onset of type II diabetes, as well as in CRC is represented by the insulin/insulin-like growth factor (IGF) system, a multifactorial signaling system that controls energy consumption, cell development, and cancer. Thus far, two insulin receptor (IR) isoforms have been identified, having distinct biological functions. IR-A primarily exerts mitogenic effects, while IR-B exerts influence on cell metabolism. The various ligand-binding abilities of the isoforms, as well as their variable tissue distribution, support this idea. IR-A is more commonly found in fetal and cancerous tissues, whereas IR-B is more commonly found in muscle, liver, and adipose tissue [16]. As reported by Vigneri et al., the PI3K-Akt pathway is up-regulated in colorectal cancer due to elevated IR-A and IGF-1R expression and IRS-1 and IRS-2 polymorphisms.

The association between DM and CRC is of particular interest, on one hand, given the prevalence of DM in the Romanian population. According to the PREDATORR study, between 1.5 and 1.9 million Romanians have DM, with the prevalence of the disease being 11.6%. Of this percentage, 2.4% were previously undiagnosed. It was observed that the prevalence of DM was higher in men than in women and increased with the advancement in age. Prediabetes, however, was found to be more prevalent (16.5%) in female patients aged between 60 and 79 [17].

On the other hand, CRC trends in Romania show a rate of hospitalization steadily increasing between 2016 and 2018, with a mortality of 34.13/100,000 in the Central and Northern regions of the country, nearly twice as high as the average range in Europe. More recent data from Globocan has estimated the new cases of CRC at 12,938 in 2020 [18].

The study aimed to verify the impact of DM on mCRC patients and to identify predictive factors relevant to the success of mCRC treatment.

## 2. Materials and Methods

### 2.1. Study Design

The present study is an observational, retrospective one, having been led in accord with the Declaration of Helsinki, approval of the Ethical Committee within the Oncohelp Medical Center being granted for the study protocol.

### 2.2. Patient Enrollment

Upon searching within the Oncohelp Medical Center patient database, 1069 individual cases of colorectal cancer were identified, based on ICD-10 medical diagnosis codes from C18 to C20. Admissions of said patients to the Oncohelp Medical Center took place between 15 June 2018 and 21 January 2022. All patients included underwent at least one cycle of chemotherapy. Chemotherapy lines initiated before 18 September 2018 were excluded from the study. Of the total number of patients, 468 cases of metastatic colorectal cancer were identified, of which 55.55% (260/478) were in male patients. The database included 8669 oncological treatment cycles related to 988 distinct chemotherapy lines. For each treatment line, the following were documented: last administration date, continuation, or completion of the line after the cut-off date (21 January 2022).

### 2.3. Statistical Analysis

A database was created, in which anthropometric data (sex, age, height, weight, body mass index (BMI)), diagnoses (diagnosis of oncological disease, associated diagnoses, histopathological examination, TNM classification, staging), hematological and biochemical determinations performed at initiation of each treatment cycle were included.

Data was collected and analyzed using SPSS v. 27 statistical software package (IBM Corp, Armonk, NY, USA).

Kolmogorov-Smirnov Normality Test was performed in order to evaluate the distribution of variables. The variables with a normal distribution are presented as mean ± standard deviation, while the non-parametric variables are presented as median (minimum, maximum). For assessing the significance of the differences between groups, we performed the t-Student test for variables with a normal distribution and the Mann-Whitney test for non-parametric variables. Time to treatment failure (TTF) was calculated as the difference between the date of the first and last administration of a treatment line (for a patient with a single administration within a single line).

For evaluating the involvement of more confounding factors (anthropometric, diagnostic, hematological, and biochemical variables) in a time-related risk for each palliative treatment line for mCRC (time to treatment failure), we built Cox proportional-hazards models using Cox Proportional Hazards Survival Regression (CPHSR).

The results were presented as risk ratios and significance levels. A *p*-value of <0.05 was considered the threshold for statistical significance.

## 3. Results

Out of 988 oncology lines, 44.12% (436/988) were administered to female patients and 55.88% (552/988) to male patients. Patients’ ages ranged from 21 to 88 years. The fact that a higher percentage of treatment lines can be observed in the male gender is primarily a consequence of those cases belonging to the 60+ age group. Within the male gender subgroup, two patients were under the age of 30. By age group, the highest prevalence of diabetes was found in men in their seventh decade of life (22%), with women reaching a similar prevalence in their eighth decade (21.3%). The distribution of patients according to gender and age groups is detailed in Table 1 and Figure 1.

Regarding the treatment options that were used, of the 988 treatment lines 84.41% (834/988) included fluoropyrimidines, 45.44% (449/988) oxaliplatin, 28.95% (286/988) irinotecan, 3.34% (33/988) trifluridine + tipiracil, 37.14% (367/988) angiogenic treatment and 20.24% (200/988) anti EGFR treatment.

In the CPHSR analysis, the presence of DM was not found to be a significant negative predictive factor for TTF. Also, no statistically significant results were obtained after evaluating DM presence as a risk factor for TTF by age-groups analysis. However, in patients included in the age group 60–69, there was a favorable predictive effect regarding the presence of DM and TTF (RR (risk ratio) = 0.81, *p* = 0.12). In the same age segment, considering only the treatment lines of the male participants, an RR of 0.74 with a p-value of 0.07 was identified. The presence of DM in male patients over 70 years of age was found to be a negative predictor with a RR = 1.66 and a *p*-value of 0.05.

Insulin treatment was present in 9.03% (14/155) of the therapy lines within the group of patients with DM and did not reach statistical significance neither as a risk nor a protective factor.

Regarding treatment with fluoropyrimidines (capecitabine and 5-fluorouracil), the maximum mean TTF was 150 days for patients with DM between 70 and 79 years of age, exceeding the maximum TFF of 132 days in patients without DM in their eighth decade of life. Furthermore, the FTT of individuals with DM indicated a clear downward trend for the age group 80 and above, a trend that was not observed in patients without DM (Table 2, Figure 2).

Oxaliplatin treatment showed a decreasing TTF duration with advancement in age in patients without DM (CPHSR revealed an RR = 1.01 for each additional year, yet no statistical significance was achieved, the value of *p* being 0.14) and a bell curve reaching a maximum in the seventh decade in people who suffered from diabetes. The average TTF was higher in patients with DM for the seventh decade, but statistical significance was not reached (Table 2, Figure 3).

In treatment lines including irinotecan, in the subgroup of patients without DM (Figure 4), the duration of TTF increased with age (CPHSR with RR = 0.99 for each additional year, without reaching statistical significance *p* = 0.41). In contrast, for patients with diabetes, the value of TTF seems to decrease with age, at least after the age of 60 (CPHSR with RR = 1.06 for each additional year, *p* = 0.06).

Treatment lines that included bevacizumab showed a shorter TTF with advancement in age for patients with diabetes over 60 years (CPHSR with RR = 1.06 for each additional year, without reaching statistical significance—*p* = 0.22, possibly due to the reduced number of treatment lines—only 44). At the same time, for patients without DM, TTF averages revealed a similarity in all age groups, with the exception of the outer segments represented by a low number of cases (Table 2, Figure 5).

Treatments that included anti-EGFR medication (Cetuximab/Panitumumab) revealed a similar age-related aspect for patients without diabetes and a bell-shaped curve for patients with DM, without indicating statistically significant correlations by age (Table 2, Figure 6).

Trifluridine/tipiracil were only included in 33 treatment lines, insufficient for the achievement of statistically valid correlations. Analysis of mean TTF values for treatments including fluoropyrimidine, oxaliplatin, or irinotecan did not reveal statistically significant differences in the *t*-test (Table 3). It has been found, however, that DM had a detrimental effect when treatments contained bevacizumab (median TTF 94 days for cases in which DM was present compared to 114 days for those patients who did not have DM, *p*-value = 0.07). Aside from the previous finding, it was also observed that those individuals with DM had a higher mean TTF when undergoing anti-EGFR therapy (median TTF 143 days for patients with DM versus 97.5 days for those without DM, having a *p*-value at *t*-test = 0.06). (Table 3)

Layered TTF analysis in relation to creatinine did not reveal statistically significant associations (Table 4, Figure 7). At the same time, the analysis by gender and presence or absence of DM revealed higher creatinine values for male patients with DM (average 0.93 mg/dl for patients with DM versus 0.84 mg/dl for those without), achieving statistical significance (*p* = 0.009) (Table 5).

A higher hemoglobin value was shown to be a favorable predictive factor for TTF (CPHSR with RR = 0.93 for each gram/dl of extra hemoglobin, *p* = 0.0005) when the whole group of patients was analyzed (916 treatment lines with available lab tests). For the treatment lines administered to DM+ patients, the protective effect was even more marked (CPHSR with RR = 0.89 for each gram/dl of extra hemoglobin, *p* = 0.02, 139 treatment lines) (Table 6, Figure 8).

The gender distribution of mean TTF values for treatments including bevacizumab did not reveal significant differences (Table 7), but additional analysis based on DM status revealed lower values of mean TTF in female patients DM+ versus DM− (81.08 days versus 193.09 days, *p* < 0.001) (Table 7). CPHSR testing of bevacizumab-treated patients (both male, and female) reveals DM+ as a risk factor for shorter TTF (RR = 1.54, *p*= 0.005). Subsequent tests further confirmed the negative predictive effect of DM+ in female patients treated with bevacizumab: RR = 2.57, *p* = 0.0001) upon CPHSR verification. The same analysis applied to men did not prove a similar impact (RR= 1.13, *p* = 0.54).

Interestingly, among the lines containing bevacizumab (*n* = 367), the combination with oxaliplatin resulted in a higher risk for those belonging to the group (RR = 1.39, *p* = 0.006), in contrast to the combination with irinotecan (RR = 1.05, *p* = 0.65).

Among the 200 treatment lines that included anti-EGFR medication, DM+ was a favorable predictive factor (RR = 0.74, but the *p*-value was 0.15). Considering all 200 treatment lines that included anti-EGFR medication, the presence of irinotecan was considered a favorable predictive factor (RR = 0.76 with *p* = 0.15).

## 4. Discussion

The present research piece represents a retrospective, observational study aimed at assessing the impact of DM on mCRC and to identify other predictive factors for the successful treatment of mCRC.

Diabetes Mellitus has consistently been shown to be an independent risk factor for colorectal cancer, with multiple meta-analyses demonstrating this premise [13,17].

Patients with DM represented 13.46% of 468 patients with metastases that underwent 988 chemotherapy treatment lines, taking into account the average age of the DM+ subgroup. The prevalence of DM in the study lot is similar to that found in the Romanian population [19]. In stage III colorectal cancer patients, the prevalence of DM was even higher (18.6% from 409 cases), the difference being statistically significant.

Other authors have shown similar DM prevalence in patients with metastatic colorectal cancer (16.3%), a reduced progression-free survival median, as well as a shorter overall survival in mCRC patients that had DM [20].

67.7% of patients without DM and 68.3% of patients with DM underwent treatment with oxaliplatin, which is surprising, given the increased risk of developing neuropathy of patients with diabetes mellitus. Furthermore, there are studies showing that hyperglycemia can negatively influence outcomes in patients with stage III CRC, leading to oxaliplatin resistance [21], inviting further research into this aspect and the investigation of other treatment options in these cases.

Regarding treatment with irinotecan, only 46.1% of patients with DM and 55.5% of those with DM had this agent included in their chemotherapy protocol. Taking into account the more favorable profile given by the addition of irinotecan to the anti-EGFR treatment in patients with DM, and the increased risk posed by the association of antiangiogenic treatment with oxaliplatin, it is worth investigating whether the use of irinotecan could be extended.

Patients with DM who received bevacizumab treatment were more likely to have comorbidities than those without DM, being in need of closer monitoring for adverse effects. Despite this factor, the OS was similar between patients with DM and those without, indicating that bevacizumab should not be excluded solely based on DM presence [22].

Regarding data provided by the complete blood panel, anemia was identified as an unfavorable predictive factor for the length of treatment of the entire lot, as well as for the sub-group of patients treated with antiangiogenic agents. These findings are in accord with data found in literature, a study published in Nature by M. Tampellini et al. underlining the utmost importance of maintaining hemoglobin values in the normal range in order for patients to benefit the most from first-line chemotherapy regimens, while anemia has been identified as a strong predictor of inferior response and survival rates in patients with advanced or metastatic colorectal cancer undergoing their first chemotherapy treatment line [23].

Among the unforeseen effects observed in the present study are the favorable impact of a higher lymphocyte count in female patients, as well as the unfavorable effect of a higher monocyte count in antiangiogenic treatment lines. In male patients, though, a higher neutrophil count had a negative predictive quality for the TTF. Similarly, other authors have found that a higher neutrophils/lymphocytes ratio (NLR) represented a negative predictive factor for PFS and OS in metastatic colorectal cancer in patients who were treated with bevacizumab and chemotherapy as a first line [24].

DM was found to be a negative predictive factor in patients undergoing treatment with antiangiogenic agents, while in those patients treated with anti-EGFR, DM was a positive predictive factor, though without reaching statistical significance (*p* = 0.07363 and *p* = 0.06219, respectively).

Among the strengths of this study, it is of note that it represents one of the first studies to investigate this topic in Romania, and the first to investigate it in the Western Region of the country. While other such studies have been conducted around the globe, one must take into consideration the particular characteristics of the Romanian population regarding the rising incidence and prevalence of DM and CRC, alongside those regarding culture, socioeconomic status, health literacy, and availability to participate in screening programs and education. Research leading to the present results was performed on a significantly large patient cohort that is very representative of the Romanian population, with a DM prevalence strikingly similar to that of the nation’s, comprising a representative pool.

A perceived limitation of the present study could be that it did not include analysis by a sub-group of non-insulin antidiabetic drugs, although the topic might be better investigated in a separate study, in order to offer further insight into the differences between the impact of these pharmacological agents as opposed to insulin on the evolution and treatment success of mCRC. Further research in this regard is already undertaken by our team.

Regarding further research possibilities, it would be worthy of investigating whether the use of irinotecan could be extended in patients with DM, given that a more favorable outcome was observed in these patients when this agent was added to the anti-EGFR treatment. Further research would also be desired relating to possible side effects of bevacizumab and oxaliplatin in patients with DM.

## 5. Conclusions

The research conducted in the present study brought forward data showing variables that influence the progression of colorectal cancer, whether the presence of DM, the efficacy of treatment lines, time to treatment failure, or progression-free survival are concerned. The favorable predictive factors identified were: the inclusion of antiangiogenic agents (entire lot), a higher hemoglobin value (entire lot, sub-group of patients with DM, sub-group treated with antiangiogenics), a higher lymphocyte count (in female patients), the inclusion of anti EGFR treatment for patients with DM, a higher creatinine value (in all treatment lines including antiangiogenic agents, as well as in the DM sub-group), and a higher lymphocyte count in treatment lines that included anti EGFR treatment.

Unfavorable predictive factors were represented by the presence of DM in female patients undergoing antiangiogenic treatment, neutrophilia in male patients, the association of oxaliplatin to treatment lines containing antiangiogenic agents, and a higher monocyte count in the aforementioned treatment lines.

## Figures and Tables

**Figure 1 medicina-58-00872-f001:**
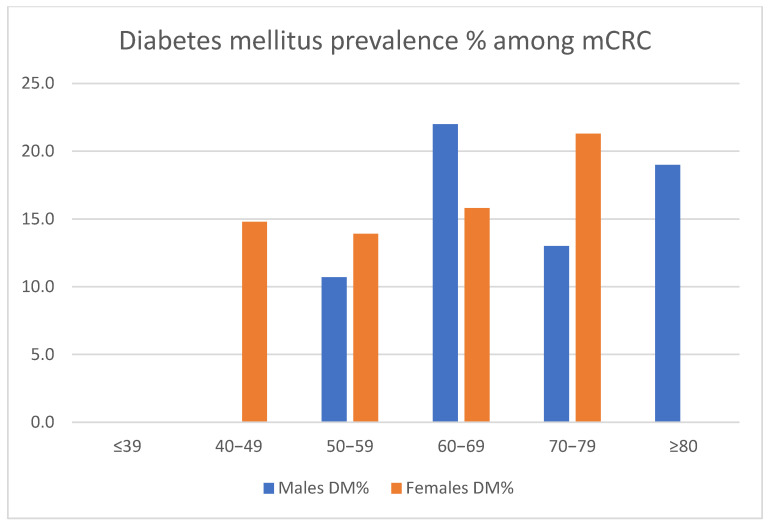
Diabetes mellitus prevalence among metastatic colorectal cancer study group (mCRC) according to age groups.

**Figure 2 medicina-58-00872-f002:**
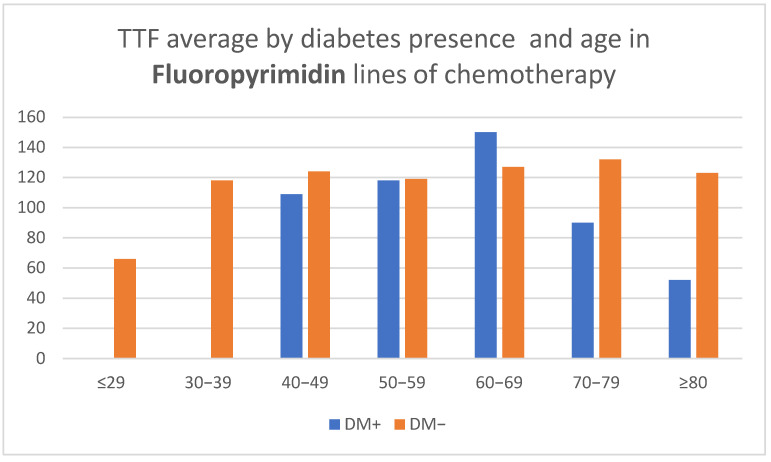
Time to treatment failure (TTF) average by diabetes presence and age in Fluoropyrimidin lines of chemotherapy.

**Figure 3 medicina-58-00872-f003:**
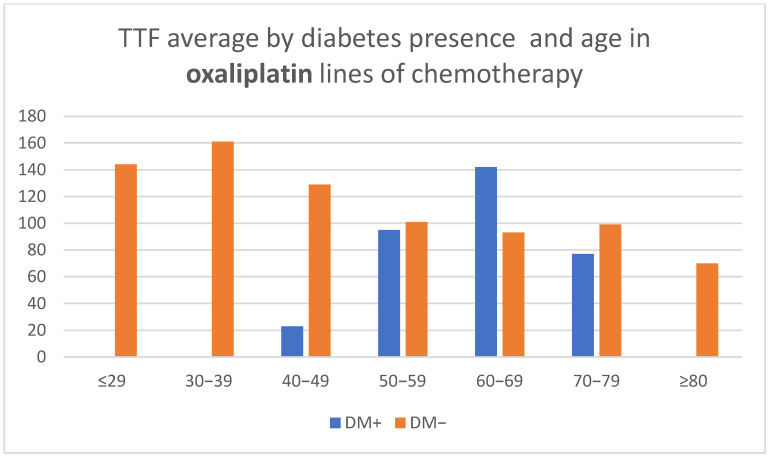
Time to treatment failure (TTF) average by diabetes presence and age in oxaliplatin lines of chemotherapy.

**Figure 4 medicina-58-00872-f004:**
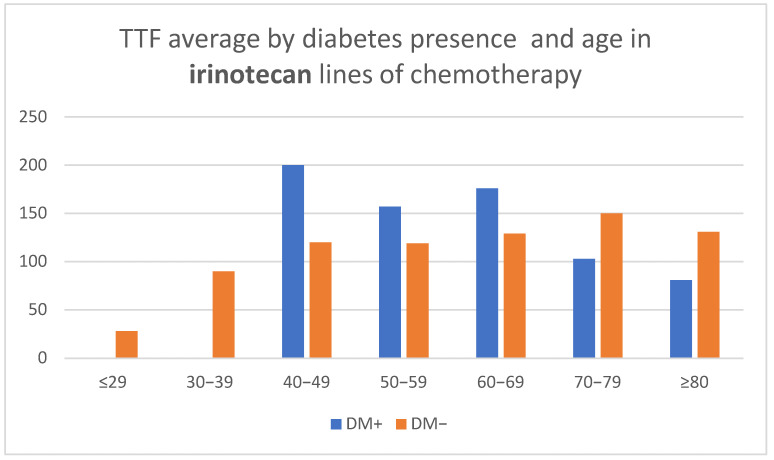
Time to treatment failure (TTF) average by diabetes presence and age in irinotecan lines of chemotherapy.

**Figure 5 medicina-58-00872-f005:**
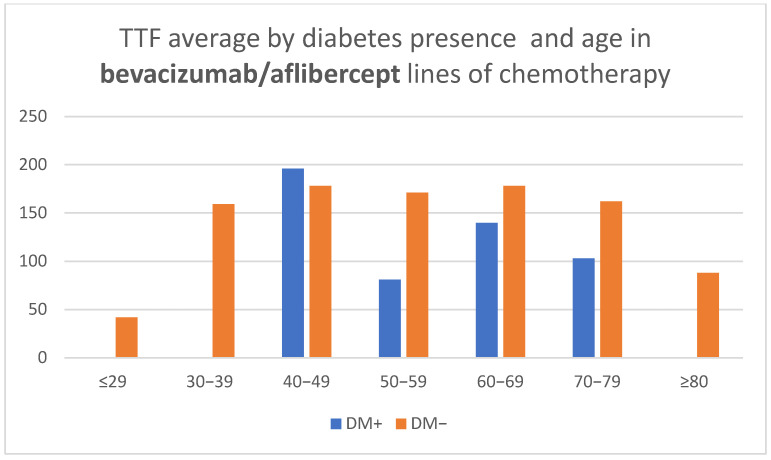
Time to treatment failure (TTF) average by diabetes presence and age in bevaci-zumab/aflibercept lines of chemotherapy.

**Figure 6 medicina-58-00872-f006:**
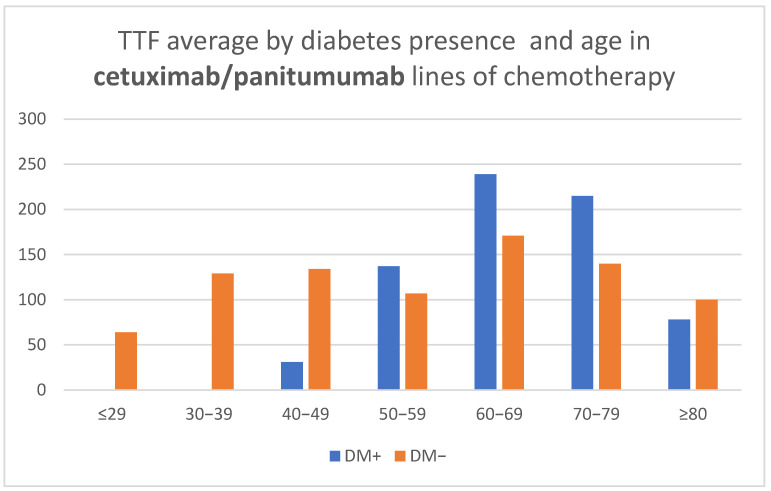
Time to treatment failure (TTF) average by diabetes presence and age in cetuximab/panitumumab lines of chemotherapy.

**Figure 7 medicina-58-00872-f007:**
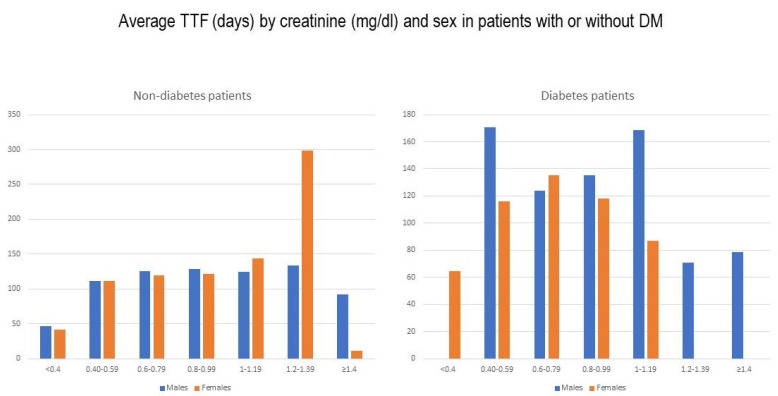
Graphic presentation of average TTF by creatinine and gender in patients with and without DM.

**Figure 8 medicina-58-00872-f008:**
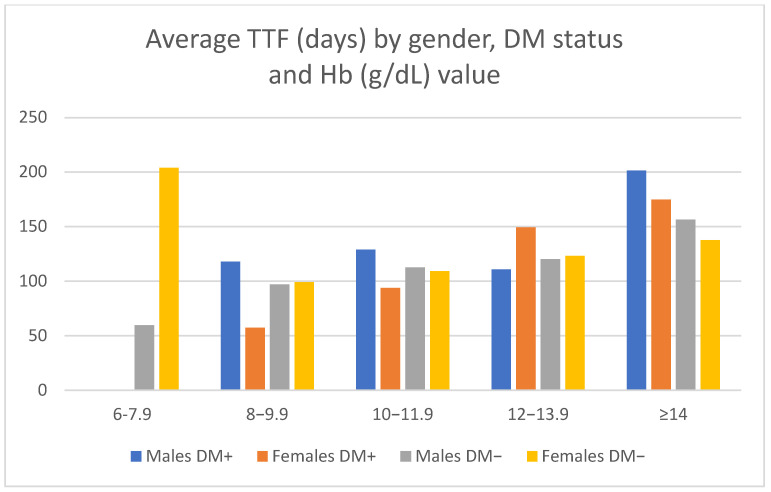
Average TTF by sex, DM status, and Hb value.

**Table 1 medicina-58-00872-t001:** Lines of chemotherapy by gender and age in patients both with and without DM.

Gender	Diabetes	Age Groups (Years)						Total
		≤39	40–49	50–59	60–69	70–79	≥80	
Male (n)	DM+	-	-	13	56	14	4	87
	DM−	11	36	109	198	94	17	465
Total male (n)		11	36	122	254	108	21	552
Male (%)	DM+	-	-	10.7	22.0	13.0	19.0	
Female (n)	DM+	-	4	17	30	17	-	68
	DM−	12	23	105	160	63	5	368
Total female (n)		12	27	122	190	80	5	436
Female (%)	DM+	-	14.8	13.9	15.8	21.3	-	

DM = Diabetes Mellitus.

**Table 2 medicina-58-00872-t002:** TTF average by diabetes presence and age in different lines of chemotherapy.

Age (Years)	≤29	30–39	40–49	50–59	60–69	70–79	≥80
Fluoropyrimidine lines of chemotherapy							
DM+ (days)	-	-	109	118	150	90	52
DM− (days)	66	118	124	119	127	132	123
Oxaliplatin lines of chemotherapy							
DM+ (days)	-	-	23	95	142	77	-
DM− (days)	144	161	129	101	93	99	70
Irinotecan lines of chemotherapy							
DM+ (days)	-	-	200	157	176	103	81
DM− (days)	28	90	120	119	129	150	131
Bevacizumab/Aflibercept lines of chemotherapy							
DM+ (days)	-	-	196	81	140	103	-
DM− (days)	42	159	178	171	178	162	88
Cetuximab/Panitumumab lines of chemotherapy							
DM+ (days)	-	-	31	137	239	215	78
DM− (days)	64	129	134	107	171	140	100

**Table 3 medicina-58-00872-t003:** Average TTF by DM presence in fluoropyrimidine, oxaliplatin, or irinotecan lines of chemotherapy.

TTF (days)	Median (IQR)	Chemotherapy Lines (n)	*p*
Fluoropyrimidine lines of chemotherapy			
DM+	87.5 (0;989)	122	0.70
DM−	76.5 (0;1168)	712	
Oxaliplatin lines of chemotherapy			
DM+	77.5 (0;761)	64	0.60
DM−	65 (0;1121)	385	
Irinotecan lines of chemotherapy			
DM+	122.5 (0;989)	42	0.42
DM−	76.5 (0;920)	244	
Bevacizumab/Aflibercept lines of chemotherapy			
DM+	94 (0;522)	56	0.07
DM−	114 (0;1168)	311	
Cetuximab/Panitumumab lines of chemotherapy			
DM+	143 (0;989)	38	0.06
DM−	97.5 (0;750)	162	

DM = diabetes mellitus; IQR = interquartile range; *p* < 0.05 indicate statistical significance.

**Table 4 medicina-58-00872-t004:** Average TTF by creatinine and gender in patients with and without DM.

Creatinine (mg/dl)	<0.4	0.40–0.59	0.6–0.79	0.8–0.99	1–1.19	1.2–1.39	≥1.4
Mean TTF (days) by creatinine and sex in DM− patients							
Male	46.25	111.33	125.64	128.7	124.62	133.11	92.63
Female	41.27	111.24	119.49	121.19	143.26	298.76	11
Mean TTF (days) by creatinine and sex in DM+ patients							
Male	-	170.67	123.67	135.14	168.54	70.75	78.75
Female	64.5	116.21	135.19	118.14	86.67	-	-

TTF = time to treatment failure; DM = diabetes mellitus.

**Table 5 medicina-58-00872-t005:** Comparison of creatinine values by gender and DM status.

Serum Creatinine (mg/dl)	Mean	STDEV	Chemotherapy Lines (n)	*p*
Female DM+	0.647	0.17	62	0.31
Female DM−	0.675	0.206	350	
Male DM+	0.933	0.258	78	0.009
Male DM−	0.844	0.237	429	

DM = diabetes mellitus; STDEV = standard deviation; *p* < 0.05 indicate statistical significance.

**Table 6 medicina-58-00872-t006:** Average TTF by sex, DM status, and haemoglobin value.

Parameter	Mean TTF (Days)				
Haemoglobin (g/dl)	6–7.9	8–9.9	10–11.9	12–13.9	≥14
Males DM+	-	117.83	128.91	110.73	201.43
Females DM+	-	57.4	93.9	149.4	174.85
Males DM−	59.67	96.93	112.68	120.24	156.38
Females DM−	204	99.15	109.17	123.18	137.7

**Table 7 medicina-58-00872-t007:** The distribution of TTF for Bevacizumab containing lines.

Factor	Median TTF (IQR)	Count	*p*
Bevacizumab containing lines TTF compared by sex			
Males	112 (0;1010)	209	0.31
Females	111.5 (0;1168)	158	
Bevacizumab containing lines TTF compared by sex and DM status			
Males DM+	138.5 (0;1168)	30	0.63
Males DM−	105 (0;1010)	179	
Females DM−	129 (0;522)	132	<0.001
Females DM+	80.5 (0;202)	26	

TTF = time to treatment failure; IQR = interquartile range; DM = diabetes mellitus; *p* < 0.05 indicates statistical significance.

## Data Availability

The collected data contains personal patient’s information. Data is available at request, after approval of the Hospital’s Ethics Committee.

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
