# Peer review of "Diabetes Mellitus and Other Predictors for the Successful Treatment of Metastatic Colorectal Cancer: A Retrospective Study"

_medicina, 2022, doi:10.3390/medicina58070872_

Round 1

Reviewer 1 Report

First of all, I would like to congratulate authors for the study and the results.

However, I would like authors to comment on some issues:

- It is a retrospective study with all the inherent biases. Do you believe that Cox model you built is appropriate- particulary in regards, that perhaps two many confounders were included.

- Moreover, I am concerned regarding the heterogenity of used CT regimens.

- in regards to DM+ groups: do you have info how well was DM medically managed?

Finally:

Could you provide some more details regarding the nature of metastatic disease (number, location of metastases)- was that taken into account as well?

Author Response

We would like to thank Reviewer 1 for the keen observations they have made and for helping us improve our manuscript as a consequence.

Cox's regression was used due to its applicability to the investigation of the effect that several variables have on the appearance of an outcome at a certain point in time, which we judged as being best. Risk factors have been independently evaluated, without iterations for multiple simultaneous factors, precisely with the purpose of excluding results derived from interdependent variables.

Regarding the heterogeneity of the CT regimens used, they were chosen by each patient's oncologist based on their clinical expertise and the particularities of the patient, which we considered to be very similar to real-life scenarios in which patients with DM receive various treatment plans. Furthermore, chemotherapy regimens available for CRC are limited to fluoropyrimidine, oxaliplatin, irinotecan and trifluridine/tipiracil, all of which were present in the study, the latter representing too few cases as to yield statistical significance upon analysis. 

DM + patients receiving treatment for cancer in the Oncohelp Clinic come from different counties, thus having records regarding DM management that are not accessible to us in Timis county. Patients being treated in the Diabetes Centre in Timis were identified, 43 having HbA1c values with a median of 7.44% at treatment initiation, of which 37 presented subsequent data regarding HbA1c (median of 7.40%). Statistical analysis could not be performed on so few patients, though future research will be conducted by our team into this subject.

The location and number of metastases were not taken into account in the present study design, they are being considered for a more detailed analysis. Regarding metastases location, 725 cases were hepatic, 348 pulmonary, 215 peritoneal, 153 lymphatic, 28 cerebral and 14 were adrenal.

Reviewer 2 Report

The relationship between DM and it's various evolutionary forms and CCR. The article analysis in a retrospective manner this relationship that is somewhat known but takes into account some parameters that don’t seem to be fully explained – time to treatment failure in patients with DM and CCR – no strong evidence seems to be present, overall survival – not properly take into account – the study is misleading – it’s not clear if the authors refer to the overall survival of CCR patients or overall survival of patients with DM (although it’s obvious it's not clearly stated).

In my mind it doses not bridge any gaps, it is well known that a patient with DM will have longer healing times, more prone to infection. DM is a manageable, not curable disease and therefore, from a certain point of view the relationship between the two doses do not bring anything different in treatment plans. The knowledge derivative from this article can be the basis for calculation predictive factors but that’s about it.

The prognostic factors that still need to be validated on a larger scale and more controlled environment (considering CCR stage, DM stage)

The manuscript requires minor English editing.

Author Response

We would like to thank Reviewer 2 for the keen observations they have made and for helping us improve our manuscript as a consequence.

Even tough time to treatment failure (TTF) analyses did not reach statistical significance, we considered it of great importance to present observations and trends, so as to provide a starting point for future research, as well as considerations for reaching statistical significance (a larger patient lot, sufficient treatment lines on which to perform analysis etc). The present study aimed to compare the duration of some classic chemotherapy lines exclusively in metastatic CRC.

The overall survival refers to patients with CCR and DM, this aspect is now clarified in the manuscript. 

While DM is an incurable disease, its management is of paramount importance for preventing acute and chronic complications, as well as for maintaining the best metabolic balance possible. It is well-known that DM has an important effect on the progression of other diseases, CRC making no exception. Numerous studies have observed that the association between CRC and DM led to a lower overall survival and a lower progression-free survival.

Regarding treatment regimens, the present study has shown that in female patients with DM undergoing bevacizumab treatment had lower TFF than those without DM. Furthermore, it was found that patients with DM undergoing bevacizumab treatment to which oxaliplatin was associated had a higher risk ratio than those to whom irinotecan was associated.

Oxaliplatin treatment presents the risk of cumulative peripheral sensory neuropathy, a fact that is o utmost importance in patients with DM that have an increased risk of developing neuropathy as a complication of the disease. Despite this fact, an important percentage of patients with DM received this treatment (68.3%), bringing forward the question of treatment election in these  patients.

Another aspect is taking into account the manner in which oxaliplatin treatment could be influenced by DM, studies showing that hyperglycemia can lead to oxaliplatin resistance.

The association between irinotecan and anti-EGFR agents should be further investigated, given the possible favorable effects observed in the present study. 

We have yet to identify studies mentioning the detrimental effect of DM on treatments including bevacizumab in female patients, an observation which could lead to the alteration of the prescribed treatment, given the existence of alternatives to bevacizumab.